# A Case Study for the Behaviors of Generalists and Specialists in Competitive Games

## Abstract

In this study, we investigate the behavioral change of a heterogeneous population as a system of information exchange. Previous approaches, such as OpenAIFive and NeuPL, have modeled a population as a single conditioned neural network to achieve rapid competitive learning. However, we found that this approach can overgeneralize the population as Generalists and hinder individual learning of specializations. To address this challenge, we propose Joint Entropy Minimization (JEM), a novel policy gradient formulation for heterogeneous populations. Our theoretical and experimental results show that JEM enables the training of Generalist populations to become Specialists. Compared to previous methods, Specialists trained with JEM exhibit increased strategy diversity, improved competitive performance, and reduced population performance disparity. These findings suggest that modeling a heterogeneous population as a group of Specialists can more fully realize the diverse potential of individual agents.

In a heterogeneous population, individuals may possess varying degrees of comparative advantage. These advantages may include physical or cognitive talents such as larger lung capacity, perfect pitch, or eidetic memory. By realizing the unique potential of individuals, the population as a whole benefits (Perez-Nieves et al., 2021). Each specialist can more competitively handle a different subset of problems. This allows individuals to focus on developing their skills and knowledge in areas where they have natural talent or ability. As a result, they can perform tasks more efficiently and effectively than others.

However, the learning representation of learning to specialize is much less explored than the well-studied learning for generalization. How the two concepts may be utilized to study the behavioral change of a heterogeneous population remains an interesting open challenge in the field of population learning.

Recent advancements in Deep Reinforcement Learning (DRL) and Game Theory-based multi-agent learning have enabled researchers to explore and simulate complex behavior change among heterogeneous agents. Two recent population studies, AlphaStar (Vinyals et al., 2019) and OpenAIFive (OpenAI et al., 2019), investigated the modeling of heterogeneous agents as a set of individual neural networks or as a single population collective in multi-agent competitive games.

In the AlphaStar research, researchers utilized human data to pretrain a set of diverse policy behaviors. Through multiple rounds of self-play, close to a thousand separate models were created to represent the diverse behavior set of the heterogeneous agents. On the other hand, OpenAIFive used a conditional neural network approach to represent a population of heterogeneous agents as a single neural network. Through distributed RL, a conditional neural network was replicated N times for each round of self-play to learn to maximize the expected return of the population.

While both methods have demonstrated empirical success in learning population competitive behaviors, two open questions remain unanswered. First, is there a way to learn the diverse behaviors of individual agents naturally without human data bias? Second, it is unclear if learning a heterogeneous population's policy behaviors as the expected return of the population can actually realize the potential of individual agent characters. A famous example is that the champion of Earthshaker in Dota 2 is known to be competitive due to its unique skill of Fissure. However, under generalized population learning, the conditional neural network of OpenAIFive was unable to realize the agent's potential.

In this paper, we investigate the long-term effects of modeling a heterogeneous population as a single Generalist model by analyzing the policy gradient integration. Our theoretical analysis enables us to predict the behavioral change of the heterogeneous population over the period of behavior optimization. Our derivation suggests that the population's behavior change over time is strongly tie to Information Theory concepts of Mutual Information and Interaction Information Maximization.

Maximizing Mutual Information between two variables in the context of two agents' actions means their shared behaviors becoming more over time. Knowing one variable gives us more information about the other. Interaction information maximization is a related concept that involves maximizing the mutual information between multiple variables. The result of population learning is similar population behaviors with a one-size-fits-all Generalist policy. To address the problem of overgeneralization, we propose a novel approach called Joint Entropy Minimization (JEM), where each agent learns to be a Specialist by maximizing their relative comparative advantage with respect to the Generalist.

Empirically, we demonstrates the effect of JEM on a diverse character population of Naruto Mobile Game, where individual character has their own unique character attributes. The game environment is in Fighting Game genre format of one versus one matches. Each character is individually incentivized to maximize its own comparative advantage to win their competitive matches. The unique characters and individualized competition environment makes it an ideal testbed for contrasting transferable skills and specialized strategies. JEM demonstrates increased diverse behaviors in the population, improved win rate competitive performances, and reduced performance disparity among agents. Our empirical results suggest that maximizing the comparative advantage of individual agents as Specialists may improve agents' abilities beyond generalization.

# 1 BACKGROUND AND RELATED WORK

Research on population learning has explored the mechanisms by which agent populations learn in both individually and as a populatio (Parker-Holder et al., 2020)(Canaan et al., 2019)(McAleer et al., 2020)(McAleer et al., 2021)(Wang et al., 2020). In these settings, the behavior of a population may evolve based on competition with opponents as well as changes in the behavior due to a population's peer influence.

## 1.1 INDIVIDUAL COMPETITIVE LEARNING

In competitive environments, the goal of multi-agent research is to optimize the performance of an agent population through competition. One approach to achieving this is individualized learning, where each agent improves its policy by learning a best-response (BR) against the policies of agents from previous iterations. The iterated elimination of dominated strategies is used to optimize a population of policies using Game Theory. Previous studies, such as (Jaderberg et al., 2018), PSRO (Lanctot et al., 2017)(Smith et al., 2021), PSRO-TRPO (Gupta et al., 2017), (Vinitsky et al., 2020), and AlphaStar, have employed various forms of self-play (Heinrich et al., 2015) to learn competitive Nash Equilibrium behaviors for a population. These approaches address issues such as stability (TRPO constraint), robustness (adversarial population), and diversity (leagues of policies pretrained on human data) in individualized learning.

## 1.2 EXCHANGE OF GRADIENT: SKILL-TRANSFER

Recent studies, such as OpenAIFive (OpenAI et al., 2019) and simplex-NeuPL (Liu et al., 2022a), have introduced a conditional population net approach for heterogeneous population learning. This approach involves replicating a conditional neural network into N copies during population self-play, with agents' IDs used to distinguish between different agents. The goal is to maximize the expected return of the population by jointly optimizing the behaviors of all agents. By communicating the learning gradient through Distributive RL, this approach has been shown to efficiently exchange individual agents' learned behavior across the population. This exchange mechanism has been more formally studied as *Skill-Transfer* (Liu et al., 2022b), where competitive behaviors are exchanged among the population.

However, we find that the mechanism of Skill-Transfer bears a striking resemblance to Mutual Information maximization, where knowledge about one variable provides more information about another. We contend that transferring common behaviors across a population may be counterintuitive to a heterogeneous population's natural diversity. Instead of leveraging the unique advantages of individual agents, Skill-Transfer may lead to overgeneralization of the population's behavior and result in poor utilization of individual agents' unique strengths.

## 2 PRELIMINARY

In this study, we consider a two-sided zero-sum game. We define a population of agents, denoted by $N = \{1, 2, \ldots, n\}$, is termed as heterogeneous if and only if each agent $i \in N$ is associated with a unique type $\alpha_i$. Each attribute is a quantifiable characteristic of the agent that influences the agent's payoff function in a game. The attributes can be either discrete or continuous variables representing features such as height, weight, skill level, etc. The game is defined by a tuple $(O, S, A, R)$, where $O$ represents the observation space and $S : O \times O$ is the joint observation of the two players in a fully observable game. $A$ represents the total action space of the heterogeneous population, where each agent has access to only a subset of $A$. The heterogeneity of the population includes individual differences in physical attributes and unique action sets for each agent. The discounted return for each player at time step $t$ defined as $R_t = \sum_{\tau=t}^{\infty} \gamma^\tau r_\tau$, where $\gamma$ is the discount factor for all agents in the range $[0, 1)$. The goal of population learning is to study the emerging behavior under different frameworks of population learning through the competition mechanism of self-play.

### 2.1 PROBLEM FORMULATION

Before analyze the mechanism of Skill-Transfer, we first lay out the premises of standard single-agent learning using Policy Gradient from (Sutton et al., 1999). We consider the learning environment as a stationary Markov decision process (MDP), where in each iteration of self-play the opponent's policy is fixed and the learner policy learns a best response (BR) in a stationary environment. At each time step $t \in \{0, 1, 2, \ldots\}$, the state, actions and rewards are denoted as $s_t \in S$, $a_t \in A$, and $r_t \in R$, respectively. To find an optimal behavior policy for an agent that maximizes reward accumulation, we model the agent's policy with a parameterized function with respect to $\theta$. The policy is characterized by $\pi_\theta(a|s) = Pr\{a_t = a|s_t = s\}$ and aims to maximize the objective function $J(\theta)$.

We denote the objective function as:

$$\text{Maximize} J(\theta) = \sum_{s \in S} d^\pi(s) \sum_{a \in A} [Adv(s, a) * \pi_\theta(a|s)] \tag{1}$$

where $d^\pi(s)$ is the stationary environment state distribution and $Adv(s, a) := Q(s, a) - V(s)$ is the advantage function representing the difference between the Q-value and the value state estimation function. The discounted return at time step $t$ is defined as $R_t^{id} = \sum_{\tau=t}^{\infty} \gamma^\tau r_\tau$, where $\gamma$ is the discount factor in the range $[0, 1)$.

From the Policy Gradient Theorem (Sutton & Barto, 2018), computing the gradient to optimize the behavior of an agent's policy can be reformulated as:

$$\nabla_\theta J(\theta) = \sum_{s \in S} d^\pi(s) \sum_{a \in A} [Adv(s, a) * \pi_\theta(a|s) * (\nabla_\theta log(\pi_\theta(a|s)))] \tag{2}$$

where we use the log trick to obtain $\nabla_\theta \log(\pi_\theta(a|s)) = \frac{\nabla_\theta \pi_\theta(a|s)}{\pi_\theta(a|s)}$.

### 2.2 EXPLORING SKILL TRANSFER

In this section, we delve into the concept of Skill Transfer, extending the single agent Policy Gradient to a conditional neural network. This exploration allows us to observe the phenomenon of Skill Transfer as a consequence of a population gradient exchange. A population gradient exchange is characterized by the influence one agent's behavioral changes have on the behaviors of other agents, given that all agents in the population share a single conditional neural network.

**Conceptual Framework:** Let's consider a basic scenario involving two distinct agents, denoted as $N := \{i, ii\}$. Under a single conditional neural network, $\{\Pi_{\theta_0}(a|s, id^k)\}_{k=i}^N$, the conditional policy maps input states and an agent's $id := \{i, ii\}$, to the respective actions of each agent. Essentially, all agents in the population share a single conditional neural network of NeuPL during the training phase. During the inference phase, the conditional neural network can emulate any agent of $id$, with $id$ serving as the conditional input.

In each distributive episode of self-play $\tau$, the learner $\{\Pi_{\theta_\tau}(a|s, id^k)\}_{k=i}^N$ learns a best response (BR) against a prioritized distribution of its past selves as the learning opponents $\{\Pi_{\theta_{0\sim\tau-1}}(a|s, id^k)\}_{k=i}^N$. We define the priority based on the Prioritized Fictional Self-Play (PFSP) (Vinyals et al., 2019) as the win rate of $\{\Pi_{\theta^{0\sim\tau-1}}(a|s, id^k)\}_{k=i}^N$ relative to the current learner.

The objective of $\{\Pi_{\theta_\tau}(a|s, id^k)\}_{k=i}^N$ can be formulated as the joint Policy Gradient to maximize the expected return of $N := \{i, ii\}$:

$$\text{Maximize } J(\theta_\tau) = \sum_{s \in S} d^\Pi(s) \sum_{a \in A} [\frac{1}{N} \sum_{k=(i,ii)} Adv(s, a^k, id^k) \tag{3}$$
$$* (\Pi_{\theta_\tau}(a^i|s, id^i) \cdot (\Pi_{\theta_\tau}(a^{ii}|s, id^{ii})]$$

Where $d^\Pi(s)$ is the opponents' prioritized distribution. $\frac{1}{N} \sum_{k=(i,ii)} Adv(s, a^k, id^k)$ is the expected Advantage estimation of the two agents, and let $\{Adv(s, a^k, id^k) = Q(s, a^k, id^k) - V(s, id^k)\}_{k=i}^N$. $(\Pi_{\theta_\tau}(a^i|s, id^i) \cdot (\Pi_{\theta_\tau}(a^{ii}|s, id^{ii})$ denotes the agents' exchange of skill or behavior.

To derive the exchange of gradient, take the derivative of Product Rule and log trick (Williams, 1992):

$$\nabla_\theta J(\theta_\tau) = \sum_{s, a^i, a^{ii}, k} Adv(s, a^k, id^k) \tag{4}$$
$$* [\, \Pi_{\theta_\tau}(a^{ii}|s, id^{ii}) \, \Pi_{\theta_\tau}(a^i|s, id^i) \, \nabla_{\theta_\tau} log(\Pi_{\theta_\tau}(a^i|s, id^i))$$
$$+ \Pi_{\theta_\tau}(a^i|s, id^i) \, \Pi_{\theta_\tau}(a^{ii}|s, id^{ii}) \, \nabla_{\theta_\tau} log(\Pi_{\theta_\tau}(a^{ii}|s, id^{ii}))]$$

Where the Advantage function may be expanded to $\{Q(s, a^k, id^k) - V(s, id^k)\}_{id=i}^N$. By applying the probability of independence and the logarithmic product rule, the exchange of gradient can be expressed as:

$$\nabla_\theta \, J(\theta_\tau) = \sum_{s, a^i, a^{ii}, k} Q(s, a^k, id^k) \left[ \Pi_{\theta_\tau}(a^i, a^{ii}|s, id^{(i,ii)}) \nabla_{\theta_\tau} \log(\Pi_{\theta_\tau}(a^i, a^{ii}|s, id^{(i,ii)})) \right] \tag{5}$$
$$- V(s, id^k) \left[ \Pi_{\theta_\tau}(a^i, a^{ii}|s, id^{(i,ii)}) \left[ \nabla_{\theta_\tau} \log(\Pi_{\theta_\tau}(a^i|s, id^i)) + \nabla_{\theta_\tau} \log(\Pi_{\theta_\tau}(a^{ii}|s, id^{ii})) \right] \right]$$

Here we separate the Value estimation's $\Pi_{\theta_\tau}^i(a^i|s)$ and $\Pi_{\theta_\tau}^{ii}(a^{ii}|s)$ as independent. This is for the reason that each joint gradient update is comparing the Advantage gained between the joint gradient of Q estimation versus each agent's current Value estimation.

### 2.2.1 POLICY GRADIENT INTEGRAL

Equation 5 demonstrates how an agent's behavior changes with each gradient pass. By performing a discrete integral $\sum$ of the duration of policy optimization, represented by $T$, over the gradient $\nabla_{\theta_\tau}$, we can utilize integration to analyze the overall change in policy behavior. Here, $T$ signifies the total duration of policy training until convergence. This approach emphasizes the use of integration as a powerful tool for understanding the evolution of policy behavior throughout the training process.

$$J(\theta_\tau) = \sum_{s,a^i,a^{ii},k} Q(s,a^k,id^k) \left[ \Pi_{\theta_\tau}(a^i,a^{ii}|s,id^{(i,ii)}) \overline{\sum_0^T} \nabla_{\theta_\tau} \log(\Pi_{\theta_\tau}(a^i,a^{ii}|s,id^{(i,ii)})) \right] \quad (6)$$

$$-V(s,id^k) \left[ \Pi_{\theta_\tau}(a^i,a^{ii}|s,id^{(i,ii)}) \overline{\sum_0^T} [\nabla_{\theta_\tau} \log(\Pi_{\theta_\tau}(a^i|s,id^i)) + \nabla_{\theta_\tau} \log(\Pi_{\theta_\tau}(a^{ii}|s,id^{ii}))] \right]$$

We can simplify the expressions by substituting $\Pi_{\theta_\tau}(a^i|s,id^i)$ with $p(x)$, $\Pi_{\theta_\tau}(a^{ii}|s,id^{ii})$ with $p(y)$ and $\Pi_{\theta_\tau}(a^i,a^{ii}|s,id^{(i,ii)})$ with $p(x,y)$. This is to replace $a^i$ as $x$, and $a^{ii}$ as $y$, we can rewrite the equation as follows:

$$J(\theta_\tau) = \sum_{s,a^i,a^{ii},k} \left[ Q(s,a^k,id^k) \left[ \mathbf{p(x,y)} log(\mathbf{p(x,y)}) \right] \right.$$
$$\left. -V(s,id^k) \left[ \mathbf{p(x,y)} \left[ log(\mathbf{p(x)}) + log(\mathbf{p(y)}) \right] \right] \right] \quad (7)$$

The simplified form of Equation 7 allows us to intuitively understand that the total behavior change of agents x and y is evaluated based on whether the joint action distribution of $\mathbf{p(x,y)} log(\mathbf{p(x,y)})$ is more advantageous than the individual estimated expected values of x and y, represented by $\mathbf{p(x,y)} [log(\mathbf{p(x)}) + log(\mathbf{p(y)})]$. The change in the action distribution, whether positively or negatively, results in the implicit maximization of their weighted Mutual Information (MI). This implies that the behavior of one agent provides information about the other, thereby demonstrating that Skill-Transfer can be mathematically expressed as the maximization of weighted MI over the behaviors of the agents.

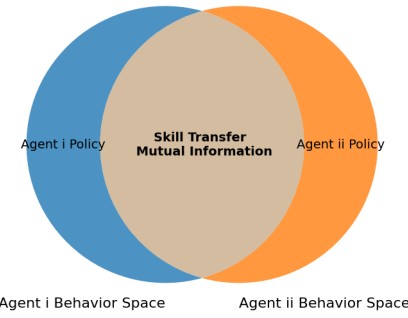

Figure 1: The figure visualizes Skill Transfer as Mutual Information Learning between two agents. Their behavior space is depicted as a Venn diagram, with common policy behaviors indicating potential competitive return. Unique, non-overlapping policy behaviors represent distinct strategies preferred by one agent but not the other.

In a population represented by a single conditional neural network, agents refine their learning process by exchanging policy gradients. This refinement is particularly focused on areas where their behaviors overlap, as shown in Figure 1. The process identifies general skills that can be applied across the agent population by maximizing their weighted Mutual Information (Guiaşu, 1977), a measure from information theory.

The weight of two probabilistic random variables, represented as $w(x,y)$, is defined as the difference $Q(s,a^k,id^k) - V(s,id^k)$ for $k \in \{i,ii\}$. This weight functions similarly to a voting mechanism, enabling individual agents to input their preferences and thereby influence the optimization trajectory of the two agents jointly. This exchange of weighted preferences improves the efficiency of the learning process, which would otherwise involve individual exploration and isolated learning.

$$I(X,Y) = \sum_{x,y} [ w(x,y) * \mathbf{p(x,y)} \log(\frac{\mathbf{p(x,y)}}{\mathbf{p(x)p(y)}})] \quad (8)$$

In general, for a heterogeneous population of size greater than two, represented as i, ii,... n, the Mutual Information takes on a more general form known as Interaction Information (InteractInfo) from Information Theory:

$$w(i;...;n+1) * I(X_i;...;X_{n+1})$$
$$= w(i;...;n+1) * [I(X_i;...;X_n) - I(X_i;...;X_n|X_{n+1})]$$

(9)

A conditional population net facilitates the exchange of gradient information across an agent population, enabling faster learning by transferring learned behavior from one agent to another. However, Skill-Transfer can also result in overgeneralization of population behaviors over time.

The Policy Gradient Integral demonstrates that over the learning duration, agents learn to maximize InteractInfo behaviors that are general and at the intersection of all agents. This concept is illustrated with a visualization of InteractInfo in Figure 2.

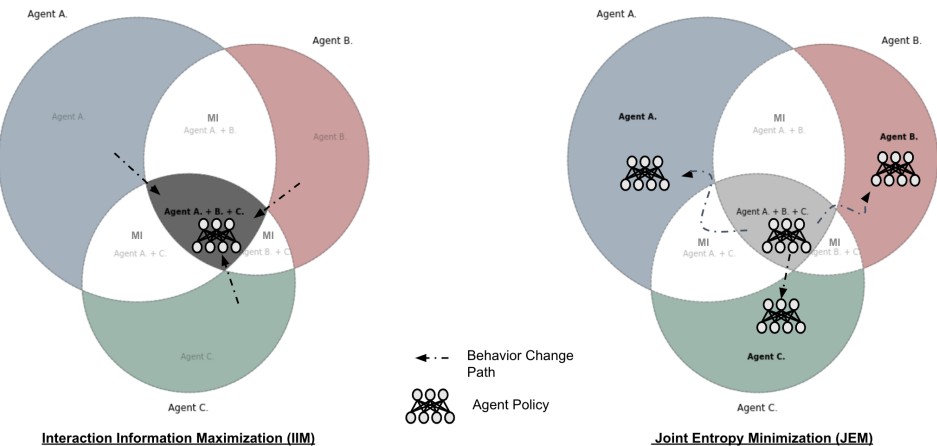

Figure 2: Figure 2 shows three heterogeneous agents (A, B, C) learning to maximize InteractInfo, resulting in a set of population behaviors represented by the central region (A + B + C). This area of behavioral intersection finds general skills transferable amongst the agents (left). However, the general skills and behaviors in the (A + B + C) region may result in a high degree of behavior similarity across the agents. Instead, we argue that to realize the full unique potential of the population, agents must learn to maximize their individual comparative advantage (right).

Figure 2 shows that when InteractInfo is maximized, the conditional population net learns a set of competitive behaviors at the intersection of the population. However, such optimization leads to a population with reduced specialization and diversity. We formally define such agent population that optimize the objective of InteractInfo maximization as the *Generalists* $\Pi_{\theta^*}(a^g|s, id)$.

## 3 METHOD

In this section, we introduce Joint Entropy Minimization (JEM) to address the problem of overgeneralization in a population's behavior. We address this problem by individualizing the advantage estimation function and policy learning for each agent.

Let the learned action behaviors of the Generalist represent the expected value of the agent population. We allow each agent to learn an individual Q-value estimation and policy BR for use in one-versus-all competitive population learning games involving N agents from the Generalist population. The goal is to help agents maximize what is unique to themselves that is better than the common skills, general behaviors exhibited by the Generalist population. The maximization process is characterized by each agent's learning of their comparative advantage relative to the expected behavior of the Generalist.

Formally, we express the maximization of comparative advantage as follows:

$$\text{Maximize} J(\psi^k) = \frac{1}{N} \sum_{g=i}^{N} \sum_{s,a^g,a^k} [Q(s, a^k, id^k) \cdot -\mathbb{H}(\mathbf{a^k}) + \mathbf{V}(\mathbf{s}, \mathbf{id^k}) \cdot \mathbb{H}(\mathbf{a^g})] \quad (10)$$

Let's define $\mathbb{H}(\mathbf{a^k})$ as the entropy of agent $k$'s action distribution, conditioned on the behavior of the Generalist. This is given by $-\pi_{\psi^k}(\mathbf{a^k})|\mathbf{a^g}, \mathbf{s}, \mathbf{id^k})\mathbf{log}(\pi_{\psi^k}(\mathbf{a^k}|\mathbf{a^g}, \mathbf{s}, \mathbf{id^k}))$. Similarly, let's define $\mathbb{H}(a^g)$ as agent $k$'s estimation of the Generalist's expected action response distribution. This is represented by the joint entropy of g = {i,ii,...n} and is given by $\pi_{\psi^k}(\mathbf{a^g})|\mathbf{s}, \mathbf{id^k})\mathbf{log}(\pi_{\psi^k}(\mathbf{a^g}|\mathbf{s}, \mathbf{id^k}))$.

The maximization of $J(\psi^k)$ in Eq 10 can be intuitively understood as the learning process of an individualized agent. This agent is learning a specific orderly action behavior, represented by $-\mathbb{H}(\mathbf{a^k})$, that aims to outperform the expected common skills or actions of the Generalist population, represented by $\mathbb{H}(a^g)$. This relative comparison is necessary for learning to specialize since "special" is a relative term anchored on what is defined as general.

## 4 EXPERIMENTS

Our research on population competitive learning was evaluated on the Fighting Game Naruto Mobile. This game allows us to test a wide range of agent behaviors without the complexity of partial observation and sparse rewards. Naruto Mobile is also widely accessible with over 100 million downloads.

We compared the performance of the "Generalists" population, optimized under NeuPL's conditional population net of up to 50 unique Naruto Mobile characters, to that of the "Specialists" population. We examined the agents' behavior through several metrics, including behavioral similarity, competitive win rates and how population size affects the overgeneralization & specialization of agents' competitive behaviors.

### 4.1 INTRODUCTION TO NARUTO MOBILE GAME MECHANICS

The game mechanics of Naruto Mobile involve real-time 1 vs 1 gameplay with a pool of over 300 unique characters, each with their own set of attributes and skills. Before the start of a match, both sides of players select their characters, auxiliary skill scroll, and pet summon. Each character has a unique set of character attributes, such as attack speed, range, and movement variations. Additionally, each character can have skills of different cooldowns, status effects, range, and duration. Things that are common among each character are the selection pool of scrolls, pet summons, and a short defensive Invincible skill to evade attacks. The key to winning a match is to reduce the opponent's health point (HP) to zero or have a higher HP at the end of a 60-second match timer.

### 4.2 BEHAVIORAL SIMILARITY

In this experiment, we aim to record and analyze the behavior similarity within the agent population. To facilitate our analysis of agent behavior, we have introduced several metrics that are specifically designed to track the *timing* of agents' skill usage.

The radial charts in Figure 3 depict the behavioral strategies of agents, categorized as Generalists (blue) and Specialists (red). The distinct variations in the red radial plot indicate that each Specialist agent employs a unique strategy during interactions, demonstrating a higher degree of behavior diversity. Conversely, the Generalists, represented in blue, show more uniform strategy variations across all agents.

In addition to visual representation, we also quantify this diversity numerically using strategy vectors. By maximizing InteractInfo and minimizing joint entropy, we optimize population behavior. The expected differences in vector distance between all pairs of agents are calculated numerically, providing a measure of similarity or dissimilarity between different members of the population.

The Generalists exhibit an expected vector distance difference of 0.9210, while the Specialists have a difference of 1.0585. This translates to a 14.9% increase in vector distance in Euclidean space for the Specialists, indicating a 14.9% increase in their behavior diversity.

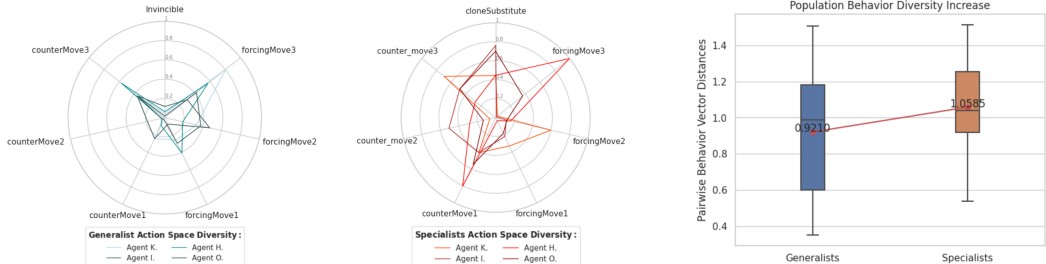

Figure 3: **Evaluation Metrics**: We assess the behavior diversity of Generalists and Specialists within the same heterogeneous character set, facing identical opponents. Behavior differences are visualized using radial plots of unit circles, where distance from the center signifies the frequency of an action, and direction represents skill utilization. Skills are categorized into forcingMove (initiating engagement), counterMove (counteracting attacks), and a substitute skill for temporary invincibility. The circle's center denotes the population's mean value, while radial edges show the maximum observed deviation.

Furthermore, we compare all pairwise agent interactions and measure the relative increase of behavior diversity on the right comparison. The higher the value, the more dissimilar two agents' actions are. This comprehensive approach allows us to analyze and compare the behavior of Generalists and Specialists effectively.

## 4.3    COMPETITIVE WIN RATES

In our second experiment, we plot the win rate performance of the top 8 Generalist and Specialist agents against the most recent policy of the Generalists. The evaluation is based on $N \times N$ competitive play among members of the agent population. Additionally we show the wall time learning curve in Appendix [**12**].

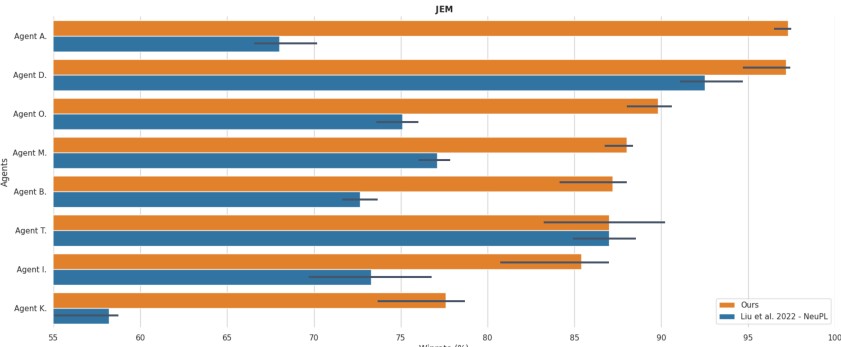

Figure 4: In the figure, we use blue to represent each agent's expected win rate as a Generalist and orange to represent their expected win rate as a Specialist. As shown in the figure, with the exception of Agent T, all other agents have experienced an improvement in their win rates after transitioning from being Generalists to being Specialists. This result suggests that most agents are capable of learning more competitive strategies that fall outside of the region of population behavior intersection.

As shown in Figure 4, there has been a general increase in the competitiveness performance of individual agents. This improvement in competitiveness performance indicates that, with the exception of Agent T's policy, other agents have become more competitive by maximizing their own comparative advantage.

In addition, our second finding on outcome disparity reveals an unconventional relationship between a population's outcome disparity and population behavior. The high degree of behavioral similarity among Generalists is associated with a high standard deviation of 9.95. In contrast, the more

unique behaviors of the Specialists have a low standard deviation of 5.99. The approximately 30% reduction reflect an improvement in a population's outcome disparity. The experiment illustrates a specialists agent population may improve both the individual agents' competitiveness as well as a more equitable outcome.

## 4.4 POPULATION SIZE STUDY

In this ablation study, we adjust the population size to carry out a comparative analysis. We have two populations: Population 1, which consists of 8 agents [I,... ,VIII], and Population 2, which expands to a population size of 50, including the initial 8 agents. We first train each population to convergence using the Simplex-NeuPL baseline (Liu et al., 2022a), resulting in two sets of Generalists (IIM8 and IIM50). We then apply JEM to enhance specialization in selected individual agents, specifically JEM $50 \Rightarrow 1$ and JEM $8 \Rightarrow 1$. The evaluation comparison is conducted based on one-vs-one matches of 1,000 games between the same agent modeled differently.

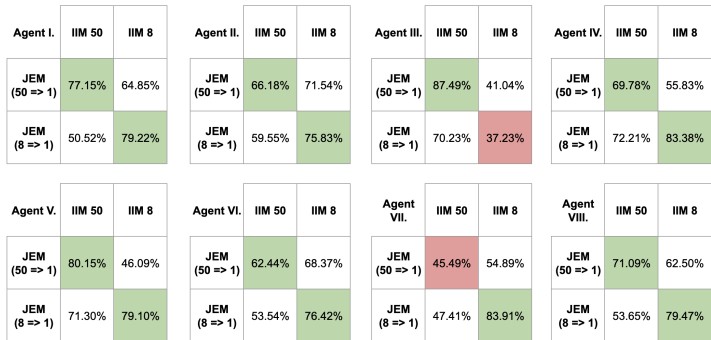

Figure 5: The figure shows the relative performance of four different population from the view of the row players. Green indicate an increase of performance from IIM training objective to JEM. Red indicate a decrease of performance.

Figure 5 illustrates that 14 out of the 16 Specialists outperforms their Generalists variation. With the two exceptions being AgentIII. (JEM $8 \Rightarrow 1$) and AgentVII. (JEM $50 \Rightarrow 1$). This indicates most agents benefit from learning to generalize to specializing with their own characteristics. In regards to the exceptions, observe that their alternative population size AgentIII. (JEM $50 \Rightarrow 1$) and AgentVII. (JEM $8 \Rightarrow 1$) shows the top 2 improvement relative to IIM. The result indicates specializing in different population size can also contributes to an agent's performance.

## 5 CONCLUSION

In this paper, we studied the impact of different Information Theory formulations on the overall behavior change of a heterogeneous population. While the popular approach of using a conditional population net can enable efficient transfer of learning across the population and increase sample efficiency, it may not always succeed in learning individual agent behavior to fully realize the potential of a heterogeneous population.

To address this, we propose JEM as a method to minimize an individual agent's action entropy relative to the average behavior of the population. Our approach enables heterogeneous agents to find more competitive behaviors than those that would be found using a set of general behaviors for the entire population. Specialization within a population not only increases the win rate performance of individual agents but also improves outcome disparity and results in a more equitable distribution of outcomes across the entire population.

**Limitation**: One limitation of our approach is the additional computation required for learning the specialization of individual agents. JEM relies on learning an individualized neural net behavior for each unique agent in a population. When the population size is large, the additional computation per agent can accumulate to a significant amount. In future research, it may be possible consider cluster the agent population into a finite set of groups.

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
