# 6 APPENDIX

# 7 MOBILE DEEP LEARNING ARCHITECTURE

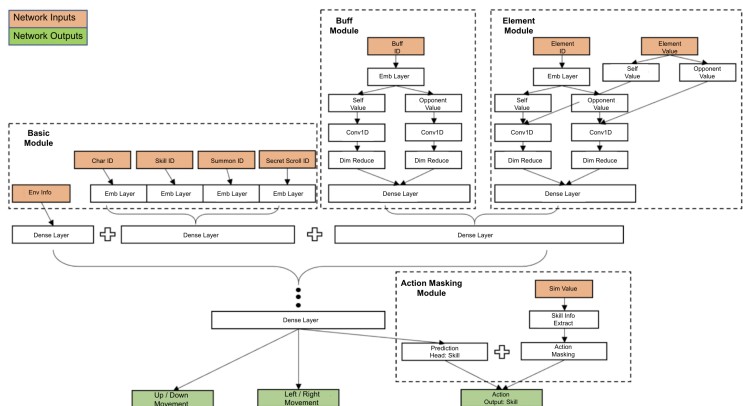

Figure 6: **DRL architecture for mobile devices:** We build our neural network deployable to mobile devices.To achieve this, we designed a neural network that utilizes an embedding layer to reduce the dimension of the inputs, and uses Conv1D instead of a Dense layer to reduce the compute load. The input to output is separated into four modules.

1. Basic Module is used for game environment statistics and characters-specific information.
2. Buff Module is used for buff and debuff information during gameplay.
3. The Element Module is used for characters' equipment, weapon IDs, scroll IDs, etc.
4. The Action Masking Module provides the model with information on whether each action is currently available.

The separation of modules in Figure 5 allows for sparse connectivity within the model, reducing compute load when compared to fully connected layers. This efficient structure allows us to perform inferences locally on mobile devices with limited computational resources.

**MDP:** The model receives input states, $s_t$, from the Basic, Buff, and Element Modules to identify characters, equipped Summons and Scrolls, characters' positions, movements, and skills' buff and element information. The model then predicts an action output, $a_t$, to control the 2D movement of the agent and the available attack and skills. The reward can be customized, but in our standard mechanics it is based on the (weighting) of an agent's own HP(10), opponent's HP(10), the result of the battle(10), combo(5), and mana(5). With the transition of the action, new state $s_{t+1}$ are given to the model for the next iteration of MDP.

## 7.1 HYPERPARAMETERS AND HARDWARE USED

2 Hyperparameters

- PPO: 0.1
- n-step: 100 frames
- Reward discount factor: 0.995
- Learning rate: 1e-4

Hardware Used

- CPUS: 5,300
- GPUS: 0

- Batch size: 300
- Compute Time:
    - $\approx 180$ Hrs

## 8 JEM POLICY GRADIENT DERIVATION

We first define a standardized Policy Gradient for a *Specialist* $k$ with Q and Value functions as:

$$\text{Maximize} J(\psi^k) = \frac{1}{N} \sum_{g=i}^{N} \mathop{\mathbb{E}}_{a^k, s, a^g} [[\, Q(s, a^k, id^k)$$
$$* \pi_{\psi^k}(a^k|s, id^k)\,] - [\, V(s, id^k) * \pi_{\psi^k}(a^k|s, id^k)\,]] \tag{11}$$

To optimize each *Specialist* $\pi_{\psi^k}$, in each step of disjoint Policy update we take the derivative of the gradient w.r.t. $\psi^k$ to isolate each policy's gradients:

$$\nabla_{\psi^k} J(\psi^k) = \frac{1}{N} \sum_{g=i}^{N} \mathop{\mathbb{E}}_{a^k s, a^g} [Q(s, a^k, id^k)$$
$$* \pi_{\psi^k}(a^k|s, id^k)\, \nabla_{\psi^k} log(\pi_{\psi^k}(a^k|s, id^k))$$
$$- V(s, id^k) * \pi_{\psi^k}(a^k|s, id^k) \nabla_{\psi^k} log(\pi_{\psi^k}(a^k|s, id^k))] \tag{12}$$

Turning the equation in to practice, we first train and define Generalist as a population of agents trained under conditional net. The training is done through the process of self-play where the current Generalist population learn to compete against all agents' past policies in competing games. The aim is to learn a set of common skills that are competitive amongst the agents. The converged competitive behaviors is also known as Nash Equilibrium (NE). It is only $\epsilon$ in approximation since DRL policies does not converge to an exact behavior. This is characterized by the floating point parameters of a neural network.

After the training of Generalist population has converged, we fixed the $\epsilon-$ NE behavior of the Generalist population as part of the stationary environment. By optimizing each *Specialist* in relation to the Generalist population, the interaction of the agents $(a^g, a^k)$ can measure the joint probability of the action output. We assume that through repeated self-play, each Specialist can approximate the action probability $a^g$ ($\epsilon$-NE Mixed strategy) as part of the environment interaction.

Given the joint probability, we rewrite the policies interaction as $\pi_{\psi^k}(a^k, id^k|a^g, s)\, \nabla_{\psi^k} log(\pi_{\psi^k}(a^k|a^g, s, id^k))$, to approximate the Joint Entropy $\approx \mathbb{H}(a^g, a^k)$. The Joint Entropy here measures the joint probability of $(a^g, a^k)$ that the *Specialist* uses the same skills and behaviors as the Generalists.

$$\nabla_{\psi^k} J(\psi^k) = \frac{1}{N} \sum_{g=i}^{N} \mathop{\mathbb{E}}_{a^k, s, a^g} [Q(s, a^k, id^k)$$
$$* \pi_{\psi^k}(a^k|a^g, s, id^k) log(\pi_{\psi^k}(a^k|a^g, s, id^k))$$
$$- V(s, id^k) * \pi_{\psi^k}(a^k|a^g, s, id^k)\, \nabla_{\psi^k} log(\pi_{\psi^k}(a^k|a^g, s, id^k)) \tag{13}$$

Below we examine the accumulated behavior change gradient $T$ over the course of optimization:

$$\text{Maximize} J(\psi^k) = \frac{1}{N} \sum_{g=i}^{N} \mathop{\mathbb{E}}_{a^k, s, a^g} [Q(s, a^k, id^k)$$
$$* \pi_{\psi^k}(a^k|a^g, s, id^k) \int_0^T \nabla_{\psi^k} log(\pi_{\psi^k}(a^k|a^g, s, id^k))$$
$$- V(s, id^k) * \pi_{\psi^k}(a^k|a^g, s, id^k) \int_0^T \nabla_{\psi^k} log(\pi_{\psi^k}(a^k|a^g, s, id^k))] \tag{14}$$

$$\text{Maximize} J(\psi^k) = \frac{1}{N} \sum_{g=i}^{N} \mathop{\mathbb{E}}_{a^k, s, a^g} [Q(s, a^k, id^k)$$
$$* \pi_{\psi^k}(\mathbf{a^k}|\mathbf{a^g}, \mathbf{s}, \mathbf{id^k}) \log(\pi_{\psi^k}(\mathbf{a^k}|\mathbf{a^g}, \mathbf{s}, \mathbf{id^k}))$$
$$- V(s, id^k) * \pi_{\psi^k}(\mathbf{a^k})|\mathbf{a^g}, \mathbf{s}, \mathbf{id^k})\log(\pi_{\psi^k}(\mathbf{a^k}|\mathbf{a^g}, \mathbf{s}, \mathbf{id^k}))] \qquad (15)$$

The behavior of a *Specialist* $k$ is initialized with high Joint Entropy with the Generalist policy due to the replication of the conditional policy. As the optimization of $\pi_{\psi^k}(a^k|s, id^k)$ begins to accumulate the updates of gradients over T, the objective of maximizing $J(\psi^k)$ converges towards minimization of the Joint Entropy, $\mathbb{H}(a^g, a^k)$, for the leading $Q(s, a^k, id^k)$. In contrast, $V(s, id^k)$ becomes the baseline policy behavior that maintains the Joint Entropy agent behavior. This minimization is achieved through the difference in the value of $Q(s, a^k, id^k) - V(s, id^k)$ on the joint action probability of $(a^g, a^k)$.

Our formulation of gradient integral allows us to analyze the resulting behavior of the individual agent's policy. Specifically, the minimization of the Joint Entropy, weighted by the Q, Value, Advantage function formulation specializes the individual agent in relation to the baseline behavior of transferable skills.

Based on the minimization effect of JEM between every *Specialist* and the $N$ *Generalists*, JEM optimizes the *Specialists* population according to each agent's heterogeneity. We refer to this transition of population behavior as Self Specialization. The benefit of specialization over generalized transferable skills is that each *Specialist* learns a unique policy that best aligns with its own characteristics. This enhances the population interactions towards diversity and boosts the performance of irregular agents by lifting the InteractInfo maximization optimization constraint.

## 9 ALGORITHM PSEUDOCODE

In this section, we present the pseudocode for our Joint Entropy Minimization (JEM) population learning approach.

For a given heterogeneous population $i, ii, ... N$ and an $\epsilon-$NE *Generalists* population $\Pi_{\theta_*}^N$, JEM optimizes $\{\pi_{\psi_0^k}\}_{k=i}^N$ to minimize the Joint Entropy of the individual agent policy against the *Generalists* population $\Pi_{\theta^*}^N$. Each one-vs-all matchmaking is sampled based on the priority given by the graph solver $F$. After a batch of episodes of NPL, each agent's policy is optimized with PPO.

---

**Algorithm 1:** JEM Multi-Agent Specialization Pseudocode

---

1 **Input**:
2 Population = {i,ii,...N} ;            // Heterogeneous population of N distinct agents
3 $\{\pi_{\psi_0^k}\}_{k=i}^N$ ;                                      // N disjoint policies
4 $\Pi_{\theta_*}^N$ ;                                        // $\epsilon-$NE *Generalists*
5 $\{\Sigma^k := (\pi_{\psi^k}, \{\Pi_{\theta^*}^g\}_{g=i}^N)\}_{k=i}^N$ ;            // Agent k's Interaction Graph
6 $F : \mathbb{R}^{1 \times N} \to \mathbb{R}^{1 \times N}$ ;         // Graph solver - NeuPL (Chen et al. 2022)
7 **Parameter**:
8 $\Pi_{\theta*}^N, \{\pi_{\psi_0^k}\}_{k=i}^N$
9 **Output**:
10 $\{\pi_{\psi_T^k}\}_{k=i}^N, \Pi^N, \{\Sigma^k\}_{k=i}^N$
11 **Algorithm Start**:
12 **for** $n \in Population$ **do**
13     $\pi_{\psi_0^n} \leftarrow \Pi_{\theta_*}^n$ ;                       // initialize N disjoint policies
14 **end**
15 **while** *(true do)* **do**
16     **for** $k \in Population$ **do**
17        $\Pi^{\Sigma^k} \leftarrow \{\Pi(a^g|s, id^g)\}_{g=i}^N$ ;        // *Specialist* k's Interaction Graph with the *Generalists*
18        $NPL(\pi_{\psi_\tau^k}, \sigma_k, \Pi^{\Sigma^k})$ ; // One-vs-all population learning with $\pi_{\psi_\tau^k}$
19        $\pi_{\psi_\tau^k} = PPO_{clip}(gradientStep(\pi_{\psi_\tau^k}))$ ;       // Optimize policy with PPO optimization
20        $U^k \leftarrow Eval(\pi_{\psi_\tau^k}, \{\Pi^g\}_{g=i}^N)$ ;        // Eval() computes the aggregate values of
21        vertex k's game outcomes. $\Sigma^k \leftarrow F(U^k)$ ;       // Define k's Interaction Graph
22     **end**
23     $\Pi^N = \Pi^N U \{\pi_{\psi_{\tau_*}^k}\}_{k=i}^N$ ;       // Adding *Specialists* to opponent pool
    ;                       // Iteratively specialize *Specialists* $\{\pi_{\psi_\tau^k}\}_{k=i}^N$.
24 **end**

---

We use an Interaction Graph, represented by $U^k$, to denote the probabilistic outcome distribution of all pairwise game matches with *Specialist* $k$. We use $F(U^k)$ to update $k$'s Interaction Graph weighted edges, which prioritizes sampling of adversarial opponents. The process is repeated until the population performance converges.

## 10   GENERALISTS SELF-PLAY ALGORITHM CONSTRUCTION

Our neural population learning for conditional population net is performed under the multi-agent Interaction Graph of NeuPL (Liu et al., 2022b). The nodes represent different generations of agents, and are connected by weighted edges $\Sigma^{(x,y)NxN}$. NeuPL provides a population self-play framework that not only competes the current population $\Pi^N\theta\tau$ with combinations of distinct agents, but also prioritizes the weighted edges for different generations of $\epsilon - NE$ population $\Pi^N\theta0 : \tau - 1$. Each population is represented as a conditional population net that learns a set of best-response (BR) strategies against all previous generations of multi-agent mixed-strategies. Each BR learns a policy that receives a return that is *epsilon* away from NE.

| Training Round | Behavior Tree | Round 1 | Round 2 | Round 3 | Round 4 | Round 5 | Round 6 | Round 7 | Round 8 | Round 9 | Round 10 | Round 11 | Round 12 | Round 13 |
|---|---|---|---|---|---|---|---|---|---|---|---|---|---|---|
| Behavior Tree | - | 0.94% | 0.21% | 0.14% | 0.22% | 0.29% | 0.23% | 0.12% | 0.12% | 0.08% | 0.07% | 0.09% | 0.06% | 0.09% |
| Round 1 | 99.06% | - | 2.56% | 1.93% | 2.12% | 2.06% | 2.57% | 2.68% | 2.01% | 2.23% | 2.41% | 2.46% | 2.01% | 1.94% |
| Round 2 | 99.79% | 97.44% | - | 4.48% | 4.66% | 5.75% | 7.08% | 6.24% | 5.44% | 5.64% | 6.13% | 6.71% | 5.11% | 4.91% |
| Round 3 | 99.86% | 98.07% | 95.52% | - | 5.91% | 6.90% | 8.86% | 9.72% | 9.18% | 10.58% | 9.95% | 10.78% | 8.29% | 8.89% |
| Round 4 | 99.78% | 97.88% | 95.34% | 94.09% | - | 14.29% | 16.56% | 21.59% | 20.24% | 20.82% | 17.84% | 18.25% | 15.69% | 15.69% |
| Round 5 | 99.71% | 97.94% | 94.25% | 93.10% | 85.71% | - | 18.70% | 21.92% | 20.82% | 21.43% | 21.56% | 21.77% | 19.78% | 19.33% |
| Round 6 | 99.77% | 97.43% | 92.92% | 91.14% | 83.44% | 81.30% | - | 17.50% | 17.33% | 18.12% | 20.03% | 20.05% | 20.05% | 19.88% |
| Round 7 | 99.88% | 97.32% | 93.76% | 90.28% | 78.41% | 78.08% | 82.50% | - | 25.02% | 25.69% | 26.21% | 26.16% | 25.68% | 24.82% |
| Round 8 | 99.88% | 97.99% | 94.56% | 90.82% | 79.76% | 79.18% | 82.67% | 74.98% | - | 33.16% | 32.85% | 31.30% | 31.24% | 31.81% |
| Round 9 | 99.92% | 97.77% | 94.36% | 89.42% | 79.18% | 78.57% | 81.88% | 74.31% | 66.84% | - | 33.87% | 33.96% | 31.71% | 33.28% |
| Round 10 | 99.91% | 97.54% | 93.29% | 89.22% | 81.75% | 78.23% | 79.95% | 73.84% | 68.70% | 66.04% | - | 41% | 32.12% | 33.89% |
| Round 11 | 99.93% | 97.59% | 93.87% | 90.05% | 82.16% | 78.44% | 79.97% | 73.79% | 67.15% | 66.13% | 59% | - | 36.10% | 36.71% |
| Round 12 | 99.94% | 97.99% | 94.89% | 91.71% | 84.31% | 80.22% | 79.95% | 74.32% | 68.76% | 68.29% | 67.88% | 63.90% | - | 42.39% |
| Round 13 | 99.91% | 98.06% | 95.09% | 91.11% | 84.31% | 80.67% | 80.12% | 75.18% | 68.19% | 66.72% | 66.11% | 63.29% | 57.61% | - |

Figure 7: The heatmap shows the evaluation matches across the different *Generalists* iterations $\Pi^N\theta t$. The ablation evaluation shows that Generalist's learning diminishes as the training approaches the 11th to 13th iteration. In particular, the 13th iteration has a win rate of only 57.6% against the 12th iteration of *Generalists*, which is close to the Nash equilibrium of 50%.

We evaluate all rounds of the *Generalists* policies against each other. In Figure 6, the heatmap shows monotonic convergence of the *Generalists* population. At the 13th iteration, the *Generalists* population has converged to an $\varepsilon-$ Nash Equilibrium, where $\varepsilon \approx 7.6\%$. Further training may only minimally increase the performance, but at the cost of reducing strategy diversity.

## 11 JEM LEARNING CURVE

This section we illustrate how the learning curves of Agents [ O, H, T, M, I, K ] differ during the learning process of JEM. At the start of the learning, each agent is equipped with a $\epsilon$ - NE Generalist conditional network. Each agent's learning process is characterized as a BR self-play against the whole agent population of $\epsilon$ - NE Generalist. The goal of each agent is to maximize the JEM objective by finding individual comparative advantage that result in higher return than the common skill of the Generalist population.

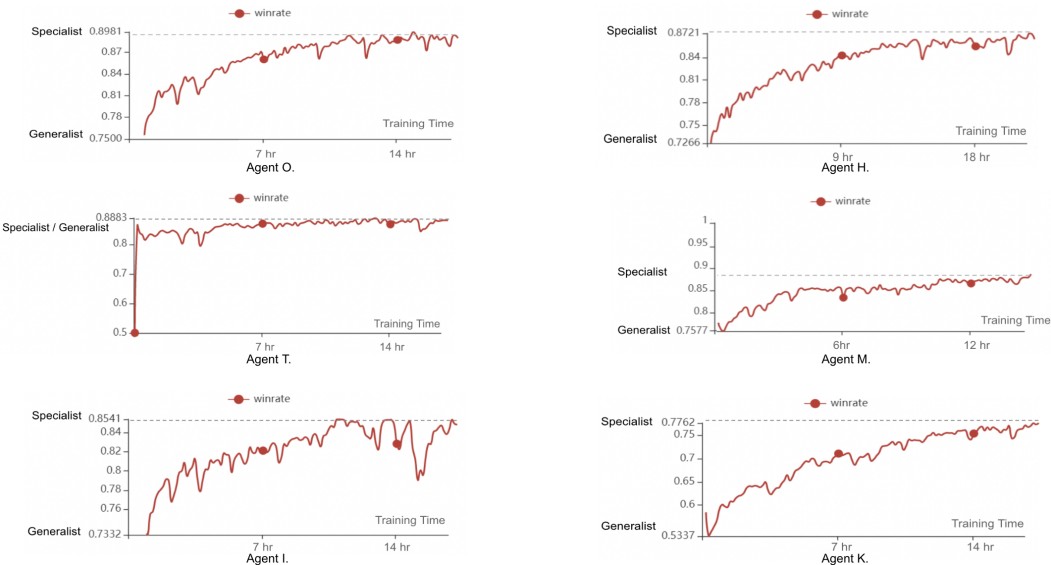

Figure 8: **Wall Time:** In the experiment we found that agents' wall time to learn to specialize can greatly vary. One of the most extreme case is Agent H., where its wall time is roughly 22 % longer than the other agents despite not showing the largest improvement. This learning curve characterizes Agent H. as a difficult agent with long time to master. At the same time, Agent T. shows little to no improvement. This special case indicate that the Generalist policy is perfectly suitable to Agent T., where learning the common skill on this agent already is the agent's preferred play style.

In Figure 5, the JEM learning curves of agents each tell a different story. There are agents with fast learning that in a short time the win rate rapidly increases as well as agents that does not increase much. The different cases indicate that some of the agents truly require special play styles to realize their potential, and the others agents their optimal strategies may lay within the common skill learned by the Generalist population. The experiment illustrates that while learning general skills that are transferrable amongst the agents may elevate the expected competitiveness of the population, learning to specialize is what make the heterogeneous population truly maximize its full potential.

### 11.1 SOFTWARE AND LICENSING

The models are implemented via Tensorflow, TensorflowLite (Abadi et al., 2015), IMPALA(Espeholt et al., 2018), and Horovod(Sergeev & Balso, 2018). These softwares are all licensed under Apache License 2.0.