# OpenReview forum: "A Case Study for the Behaviors of Generalists and Specialists in Competitive Games"
_ICLR.cc/2024/Conference — Submitted to ICLR 2024_

### Official Review · Reviewer_XoZX · 2023-10-15

**Soundness:** 2 fair
**Presentation:** 1 poor
**Contribution:** 2 fair
**Rating:** 3
**Confidence:** 2

**Summary:**

The authors propose a method, JEM, to increase the strategy diversity in population-based training in the hope that such a method would improve the performance. Empirical tests suggest that their method indeed improves upon the state of the art.

**Strengths:**

The method seems novel and interesting, and seems to exhibit favorable experimental results over the state of the art.

**Weaknesses:**

I found the paper impossible to follow, and gave up on parsing it partway through. Perhaps this is in part due to my lacking certain pieces of background, but I do feel like I am at least somewhat in the target audience of this paper. Here are a few writing concerns, in no particular order.

1. Many technical terms are not defined, some of which are fundamental to understanding the paper. These things ought to have formal, mathematical definitions considering their importance to the paper. Some examples:

    a. Sec 2.1: What is a "population", and what does it mean for a population to be "heterogeneous"?

    b. Sec 2.2: What is "gradient exchange"?

    c. Sec 2.2.1: I understand x and y to stand in for a^i and a^ii respectively. Is this accurate? In any case this should be stated.

    d. Sec 3: "Generalist" should be defined formally

1. Should I be considering a partially-observable game, or a fully-observable game? The prelims suggest partial observability, but e.g. policies are conditioned on states, which suggests full observability
1. My understanding is that the setting is a two-player zero-sum game. In that setting, a player should never correlate its action with the opposing player, i.e., the mutual information (8) should always be 0 in equilibrium. Why, then, is it interesting to consider (or, indeed, to optimize) the mutual information?

The experimental results seem reasonable, but my ability to parse them is essentially limited to "I see figures in which the current paper seems to do well".

Due to the writing issues alone, I don't think this paper is publishable in the current state. I vote to reject. If the authors could perform a revision that improves the quality of writing and explanations, I would read such a revision and may change my score.

Nitpicks (not affecting evaluation):

1. $*$ should probably not be used as a generic multiplication symbol---prefer $\cdot$ (or simply juxtaposition)
1. Some brackets are improperly sized, e.g., Eq (3)
1. Sec 2.2.1: Shouldn't the integral be actually a sum, since at the end of the day these training algorithms operate in discrete time?
1. $\log$ should always use the backslash, i.e., not $log$.
1. It's a bit confusing to use Roman numerals to index the players; why not just use standard Arabic numerals (1, 2, ..., N), as is consistent with most game theory literature?
1. Preliminaries typo: $S : O \times O$ doesn't parse
1. The paper contains a large number of grammatical errors, too many to enumerate. Proofreading would go a long way.
1. Eq (10): it should be explciitly stated that $\mathbb H()$ is the joint entropy (assuming that it actually is).

**Questions:**

Some questions are listed in the above section; I have none beyond that.

---

> ### Author Response · Authors · 2023-11-15
>
> We are delighted that the reviewer find our method novel and interesting, and that it exhibits favorable experimental results over the state of the art. We appreciate the reviewer's feedback and have revised our paper to address the concerns.
>
> $\textbf{Q1:}$ What is a “population”, and what does it mean for a population to be “heterogeneous”?
>
> $\textbf{A1:}$ A population refers to a set of agents that are trained simultaneously in a population-based training framework. A population is considered heterogeneous if the agents have different attributes, such as height, weight, etc.
>
> $\textbf{Q2:}$ What is “gradient exchange”?
>
> $\textbf{A2:}$ Gradient exchange is a technique that allows agents to share their gradients with each other during training. This can facilitate exploration, diversity, and coordination among the agents.
>
> $\textbf{Q3:}$ I understand x and y to stand in for a^i and a^ii respectively. Is this accurate? In any case this should be stated.
>
> $\textbf{A3:}$ Yes, the reviewer is correct. We will make clarification in paper's revision.
>
> $\textbf{Q4:}$ “Generalist” should be defined formally
>
> $\textbf{A4:}$ A generalist is a policy that controls a population of agents that the training objective is such that the policy can be applied to a heterogeneous agent population. We have added a formal definition of generalist in the paper.
>
> $\textbf{Q5:}$ Should I be considering a partially-observable game, or a fully-observable game? The prelims suggest partial observability, but e.g. policies are conditioned on states, which suggests full observability
>
> $\textbf{A5:}$ We consider both partially-observable and fully-observable games in our paper. During competition, individual agents only observe their individual observations. After a competition, for example rock paper scissors, all agents see what the other agents have done. Liu et al. 2022's proposed condition neural net of NeuPL takes advantage of this by letting all agents jointly optimize on all agent’s experience. In other words, during training the fully observable experiences are used to train all agents. We have made this distinction more clear in the paper.
>
> $\textbf{Q6:}$ My understanding is that the setting is a two-player zero-sum game. In that setting, a player should never correlate its action with the opposing player, i.e., the mutual information (8) should always be 0 in equilibrium. Why, then, is it interesting to consider (or, indeed, to optimize) the mutual information?
>
> $\textbf{A6:} We agree that in a two-player zero-sum game, the mutual information between the actions of the players should be zero in equilibrium. However, we are interested in the problem setting of population learning via self-play and policy gradient optimization. In this context, Eq. (3) is the result of (x and y) versus (x’ and y’), where (x’ and y’) are (x and y)’s opponent. As proposed by Liu et al 2022, NeuPL can make the two policies’ optimization more efficient by jointly optimizing them under a single conditional policy (conditioned on x and y’s id). The neural architecture of NeuPL is a single conditioned policy that jointly optimizes a population of agents. In Liu et al 2022’s paper, they showed that there is a Skill Transfer effect that makes agents’ learning more efficient. Our paper also finds the training would be more efficient, but at NeuPL’s convergence, it would not be optimal. Optimal here means there is still room for improvement for the individual agents to further improve.
>
> We are making the necessary changes in our paper this week. The revision will be underlined in red, to address the above symbols and provide a clearer understanding of our work. We value your feedback and welcome any further questions. If our answers is satisfactory, we kindly ask to consider increasing the score of our paper.

---

> > ### Comment · Reviewer_XoZX · 2023-11-15
> > **Response - still need more formality**
> >
> > The definitions given by the authors still lack formalism. For example: you say that "A population is considered heterogeneous if the agents have different attributes, such as height, weight, etc.", but I don't know what an "attribute" of an agent is in a general (PO)MDP.
> >
> > Here's an example of a formal definition (just making up symbols. This is probably not close to what you actually want to define, but it's just to give a sense of what I mean by *formal*)
> >
> > **Definition:** A collection of training algorithms (one per agent) is called *heterogeneous* if the policy played by each agent depends only on the training data (observations, actions, rewards) observed and played by that agent.
> >
> > For example, this definition would rule out training algorithms involving centralized control/advice, weight sharing, etc.
> >
> > (By the way, I'd normally use the word "uncoupled" to describe my above definition, so in the small chance that the above is actually the definition you want, I'd advise switching terms to be more in line with standard language in game theory)
> >
> > The informal, plain-language definitions given by the authors are good for intuition, but in a technical paper I would expect more formalism along the above lines, especially for notions that are new to this paper.
> >
> > I will keep my score. I recommend a major revision in which the authors formalize important definitions along the lines that I have mentioned.

---

> > > ### Author Response · Authors · 2023-11-17
> > >
> > > We are thankful for your constructive feedback, and understand your decision of keeping your score.
> > >
> > > The paper will be revised and satisfy your critiria next time if we are lucky to have you as a reviewer again.
> > >
> > > Appreciative.

---

### Official Review · Reviewer_prT3 · 2023-10-30

**Soundness:** 2 fair
**Presentation:** 1 poor
**Contribution:** 2 fair
**Rating:** 3
**Confidence:** 4

**Summary:**

The paper investigates how to train heterogeneous agents in competitive games to learn specialized behaviors that leverage their unique strengths, rather than overgeneralizing to a common set of general skills. The paper proposes a novel policy gradient method called Joint Entropy Minimization (JEM), which minimizes the joint entropy between an agent's actions and the average actions of the population. The paper evaluates JEM on a fighting game environment with diverse character attributes and skills and shows that JEM leads to increased behavior diversity, improved win rates, and reduced outcome disparity among agents.

**Strengths:**

- The paper addresses an interesting and relevant problem of learning heterogeneous behaviors for a diverse population of agents in competitive games.
- The paper evaluates JEM on a realistic and challenging game environment, Naruto Mobile, with over 300 unique characters and complex game mechanics. The paper uses several metrics to measure the behavior similarity, competitive win rate, and outcome disparity of the agents.

**Weaknesses:**

- The writing of this paper is very unclear and the notation is not precise so it is very hard to follow what is the exact setting and main contribution of the paper.
- The code is not provided for reproduction.

**Questions:**

- In the PRELIMINARY section, the problem is set as a zero-sum game, why in Eq. (3) the problem is for two policies to jointly maximize a single objective?
- Also in Eq. (3), why are two policies share the same parameter \theta_\tau? Shouldn’t they have their own different parameters?
- In the first line of PRELIMINARY, N={1,2,…,N} is bad notation.
- Right after Eq. (1), the definition of Q(s,a) and V(s, id) are the same?
- Also in this line, what is the definition of id? It is the first time it appears but without properly defined.
- In Eq. (6), what is the definition of T? What is the policy gradient integral over?

---

> ### Author Response · Authors · 2023-11-15
>
> We are delighted that the reviewer recognizes the relevance and interest of our paper’s problem, which investigates how to train heterogeneous agents in competitive games to learn specialized behaviors. We appreciate the reviewer’s acknowledgement of our evaluation of JEM on the challenging game environment, Naruto Mobile, with its diverse character attributes and skills.
> We apologize for the unclear writing and notation in the paper. We will revise the paper to improve the clarity and precision of the presentation. We will also provide the code for reproduction upon acceptance.
> We thank the reviewer for the questions and we answer them as follows:
>
> Q1: In the PRELIMINARY section, the problem is set as a zero-sum game, why in Eq. (3) the problem is for two policies to jointly maximize a single objective?
>
> A1: The problem setting we are tackling is that of population learning via self-play and policy gradient optimization. In this context, Eq. (3) is the result of (x and y) versus (x’ and y’), where (x’ and y’) are (x and y)’s opponent. As proposed by Liu et al 2022, NeuPL can make the two policies’ optimization more efficient by jointly optimizing them under a single conditional policy (conditioned on x and y’s id).
>
> Q2: Also in Eq. (3), why do two policies share the same parameter \theta_\tau? Shouldn’t they have their own different parameters?
>
> A2: The neural architecture of NeuPL is a single conditioned policy that jointly optimizes a population of agents. In Liu et al 2022’s paper, they showed that there is a Skill Transfer effect that makes agents’ learning more efficient. Our paper also finds the training would be more efficient, but at NeuPL’s convergence, it would not be optimal. Optimal here means there is still room for improvement for the individual agents to further improve.
>
> Q3: In the first line of the PRELIMINARY, N={1,2,…,N} is bad notation.
>
> A3: We agree that this notation is confusing and we will change it to N={1,2,…,n}.
>
> Q4: Right after Eq. (1), the definition of Q(s,a) and V(s, id) are the same?
>
> A4: We agree that the definition ought to be different. Q(s,a) is the state-action value function, which is the expected return starting from state s and taking action a. V(s, id) is the state value function for agent id, which is the expected return starting from state s and following the policy of agent id. We will make this distinction more clear in the paper.
>
> Q5: Also in this line, what is the definition of id? It is the first time it appears but without being properly defined.
>
> A5: id is the agent index, which ranges from 1 to n. We will define it before using it in the paper.
>
> Q6: what is the definition of T? What is the policy gradient integral over?
>
> A6: T is the total duration of training. Policy gradient integral over the change of parameter (over the training duration).
>
> We have made the necessary changes in our paper with red line highlighting the changes. We value the reviewer's feedback and welcome any further questions. If our answers is satisfactory, we kindly ask to consider increasing the score of our paper.

---

### Official Review · Reviewer_h8i4 · 2023-11-01

**Soundness:** 3 good
**Presentation:** 2 fair
**Contribution:** 3 good
**Rating:** 5
**Confidence:** 2

**Summary:**

This paper studies the behavioural change of a heterogeneous population, and shows that existing approaches may overgeneralize the population as Generalists and hinder individual learning of specialization. A new method based on joint entropy maximization is proposed. The proposed method is shown to increase behavioural diversity and reduce performance disparity.

**Strengths:**

1. The key research questions are well-motivated.
2. The theoretical analysis of existing approaches, which shows the connection to mutual information, provides new insights.
3. The proposed solution is reasonable.
4. The paper is generally well-written.

**Weaknesses:**

Some important technical details about this paper lack clarity and need to be further clarified:
1. While one of the main results, Equation 7, looks like mutual information, what does the term $w(x,y)$ of mutual information (defined in Equation 8) correspond to? An explicit connection to mutual information right after Equation 7 is helpful for understanding the claim "The exchange of Policy Gradient between two agents results in the implicit maximization of their Mutual Information (MI) over time. "
2. Similarly, there seems to be a typo in Equation 10 (another key result). In addition, the terms in this equation have not been explained immediately following this equation, making it hard to understand the intuition.
3.  While the paper has emphasized the disadvantage of "generalist", the proposed method (at least, Equation 10) still depends on the policy of the "generalist."  Does the proposed method rely on the existing method to learn the policy of the "generalist"?  How does the proposed method strike a balance between being a "generalist" and being a "specialist"? Will this matter in certain scenarios? The discussion on these issues is not adequate.

**Questions:**

I would like to see responses to the above weaknesses.

---

> ### Author Response · Authors · 2023-11-15
> **Discussion of Weighted Mutual Information And Specialists Versus Generalists**
>
> We are delighted that the reviewer recognizes the strengths of our paper, including the well-motivated research questions, insightful theoretical analysis, and the reasonableness of the proposed solution. We appreciate the constructive feedback and have made revisions to our paper accordingly. Here are our responses to the specific questions:
>
> Q1: What does the term w(x,y) of mutual information (defined in Equation 8) correspond to?
>
> A1: The term w(x,y) of mutual information is first introduced by Guiasu 1977 in “INFORMATION THEORY AND A STOCHASTIC MODEL FOR EVOLUTION”. It is defined as the expected weighting over a set of random variables. In the context of Multiagent Policy Gradient, this expected weighting corresponds to the expected advantage function (Q-V) for all agents, jointly optimized under a single conditional neural net. This denotes the weighting of agents x and y on whether a policy change towards Q leads to a better policy than the existing estimated expected value of V. We have added this explanation immediately after Equation 7 in the revised paper for clarity.
>
> Q2: There seems to be a typo in Equation 10. Could you clarify the terms in this equation?
>
> A2: We apologize for the typo in Equation 10. The * has been replaced with a dot product symbol ($\centerdot$) in the revised paper. Equation 10 illustrates agent k's optimization of behavior to gain an advantage relative to a population of Generalists. The term $V(s,id^{k}) \centerdot \mathop{\mathbb{H}}(a^{g}, a^{k})$ represents the relation of agent k's expected value and the entropy of the joint action distribution of agent k and the Generalists. The term $Q(s,a^{k},id^{k}) \centerdot - \mathbf{\mathop{\mathbb{H}}(a^{g}, a^{k})$ represents the expected value of agent k's specific state and action, weighted by the negative entropy of the joint action distribution. We have added an explanation of these terms immediately after Equation 10 in the revised paper.
>
> Q3: Does the proposed method rely on the existing method to learn the policy of the "generalist"? How does the proposed method strike a balance between being a "generalist" and being a "specialist"?
>
> A3: The proposed method does rely on the existing method to learn the policy of the "generalist". The policy of the "generalist" is learned by optimizing a set of population behavior that would be, in general, competitive for the population. Our approach of JEM optimizes agents’ individual’s policies as a population of "specialists". To optimize the agent’s policy gradient not based on what would be optimal for the general population, but what would be advantageous for each agent to deviate from the population norm (As “special” is a relative term to the term what is considered “general”). The proposed method aims to fully optimize agent’s individual potential rather than striking a balance as the case study we performed is in the scenario of noncooperative game. The goal would be to enable each agent to achieve individual optimality amongst the population.
>
> We have made the necessary changes in our paper with red line highlighting the changes. We value the reviewer's feedback and welcome any further questions. If our answers is satisfactory, we kindly ask to consider increasing the score of our paper.

---

> > ### Comment · Reviewer_h8i4 · 2023-11-22
> >
> > Thank you for your detailed response. I think my third question has not been properly addressed. It doesn't make sense to me why striking a balance between "generalist" and "specialist" does not matter, since the proposed method relies on "generalist". The "generalist" and "specialist" seem to be coupled together in the proposed method.

---

### Official Review · Reviewer_14MZ · 2023-11-01

**Soundness:** 2 fair
**Presentation:** 1 poor
**Contribution:** 2 fair
**Rating:** 3
**Confidence:** 3

**Summary:**

This paper studies how a heterogeneous group of agents learn while interacting in a competitive way. More precisely, the authors focus on the question of specialization. They propose a way to make agents who start as generalists learn how to specialize.

**Strengths:**

The paper considers an interesting question. The tools used (based on information theory) also seem relevant.

**Weaknesses:**

I found the paper a bit hard to read, partly due some typos or symbols that are not (or not well enough) defined or explained.

**Questions:**

1. Page 3: below (1) what is $id$ in the inputs of $V$?

2. Page 4, line 3: Is $N$ a \emph{set} composed of two elements which represent the indices of two players? If so, why is there $\{\dots\}_{k=i}^N$ in the following line?

3. Page 5, equation (9): Could you please give or recall the definition of $w(i;\dots;N+1)$?

4. Page 6, definition of $J$: Could you please clarify how $\psi^k$ in the left-hand side affects the right-hand side? Also, is the $*$ a typo and, if not, what is its meaning?

---

> ### Author Response · Authors · 2023-11-15
> **Discussion QnA of Notations**
>
> We are delighted to hear that the reviewer find our paper interesting and the use of tools based on information theory relevant. We thank the reviewer for their interest and constructive feedback on our paper. We have revised our paper accordingly and addressed the questions as follows:
>
> Q1: On Page 3, what is 'id' in the inputs of V(s,id)?
>
> A1: The 'id' input is a variable that indicates the identity of a game player. For example, in a game of chess, if a player moves first, 'id' may be set to 0 and to 1 if the player moves second. This 'id' input is a key feature that enables the NeuPL model architecture, where each agent in a population has an 'id' to compete in an environment.
>
> Q2: On Page 4, is 'N' a set composed of two elements which represent the indices of two players? If so, why is there 'k = i...N' in the following line?
>
> A2: Yes, the reviewer is correct. 'N' is a set of two elements that represent the indices of two players. This is initially used to examine Skill Transfer when there are 2 agents trained under one conditional neural net. We wrote 'k=1...N' because we later extend the analysis to more than 2 agents, where 'N' could be any integer greater than 2.
>
> Q3: On Page 5, equation (9), could you please give or recall the definition of 'w'?
>
> A3: The weight of the Interaction Information term 'w' is defined as the expected weighting over a set of random variables. In the context of Multiagent Policy Gradient, this expected weighting would be the expected advantage function of (Q-V) for all agents jointly optimized under a single conditional neural net.
>
> Q4: On Page 6, definition of '$\psi^k$', could you please clarify how '$\psi^k$' in the left-hand side affects the right-hand side? Also, is the '\psi' a typo and, if not, what is its meaning?
>
> A4: The term '$\psi^k$' influences the right-hand side by altering the policy action distribution of '$a^k$', in terms of the action probability distribution entropy $\mathbb{H}$. The full equation is quite extensive, but its explicit form is shown in equation 15 as '$\pi_{\psi^k}(a^k|a^g,s,id^k) log(\pi_{\psi^k}(a^k|a^g,s,id^k)$'. The * in the right-hand side is a multiplication sign. We have changed it to $\centerdot$ as dot product in the paper revision to make it clearer.
>
> We have made the necessary changes in our paper with red line highlighting the changes. We value the reviewer's feedback and welcome any further questions. If our answers is satisfactory, we kindly ask to consider increasing the score of our paper.

---

### Meta-Review · Area_Chair_GuEk · 2023-12-07

**Metareview:**

a) Claims: This paper proposes a procedure that aims to train agents in a competitive setting to "specialize" in their own particular strengths, rather than "overgeneralizing" by learning a general, common set of skills.

b) Strengths: There was broad agreement that the question was interesting, and that the empirical evaluation seemed positive.

c) Weaknesses: The reviewers all agreed that the paper had clarity issues that make it impossible to accept in its current form.  The key concepts and results need to be described formally and precisely.

**Justification For Why Not Higher Score:**

The paper's clarity does not meet the bar for publication at ICLR.

**Justification For Why Not Lower Score:**

n/a

---

### Decision · Program_Chairs · 2024-01-16

Reject